# Assessment of content validity and psychometric properties of VISA-A for Achilles tendinopathy

**Jonathan Comins** [1,2], **Volkert Siersma**[1], **Christian Couppe**[3,4], **Rene B. Svensson**[3], **Finn Johansen**[3], **Nikolaj M. Malmgaard-Clausen**[3], **S. Peter Magnusson**[3,4]*

**1** The Research Unit for General Practice and Section of General Practice, Department of Public Health, University of Copenhagen, Copenhagen, Denmark, **2** Section for Sports Traumatology M51, Bispebjerg and Frederiksberg Hospital, University of Copenhagen, Copenhagen, Denmark, **3** Institute of Sports Medicine Copenhagen, Bispebjerg Hospital, and Center for Healthy Aging, University of Copenhagen, Copenhagen, Denmark, **4** Department of Physical and Occupational Therapy, Bispebjerg Hospital, Copenhagen, Denmark

* p.magnusson@sund.ku.dk

**Data Availability Statement:** All relevant data are within the paper and its Supporting Information files.

## Abstract

A recent COSMIN review found that the Victorian Institute of Sports Assessment–Achilles tendinopathy questionnaire (VISA-A) has flawed construct validity. The objective of the current study was to assess specifically the process of how VISA-A was constructed and validated, and whether the Danish version of VISA-A is a valid patient-reported outcome measure (PROM) for measuring the perceived impact of Achilles tendinopathy. The original item generation strategy for content validity and the process for confirming the scaling properties (construct validity) were examined. In addition, construct validity was evaluated directly using several psychometric methods (Rasch analysis, confirmatory factor analysis (CFA), and multivariable linear regression) in a cohort of 318 persons with Achilles tendinopathy with symptom duration groups ranging from less than 3 months to more than 1 year of chronicity, and a group of 120 healthy persons. We found that the item generation and item reduction in the original construction of VISA-A was based on literature review and clinician consensus with little or no patient involvement. We determined that 1) VISA-A consists of ambiguous conceptual item themes and thus lacks content validity, 2) there was no thorough investigation of the psychometric properties of the original version of VISA-A, which thus lacks construct validity, and 3) rigorous direct assessment of the psychometric properties of the Danish VISA-A revealed inadequate psychometric properties. In agreement with the COSMIN study, we conclude that when used as a single score, VISA-A is not an adequate scale for measuring self-reported impact of Achilles tendinopathy.

## Introduction

The Victorian Institute of Sports Assessment–Achilles tendinopathy questionnaire (VISA-A) [1] is the most widely used patient-reported outcome measure (PROM) for studies of Achilles tendinopathy [2]. However, a recent editorial questions the usefulness of VISA-A, and

**Funding:** The authors received no specific funding for this work.

**Competing interests:** The authors have declared that no competing interests exist.

underscores the limited evidence of its "clinimetric properties" [3]. Furthermore, a recent COSMIN checklist review concluded that all 11 versions of VISA-A have flawed construct validity and weak responsiveness [4].

The purpose of PROMs is to measure a person's perception of the impact of some pathology, not the physical pathology itself [5]. The measurement process consists of assigning numerical values to the responses to the questions (items). These values are summed in an overall (total) score that should represent the magnitude of the problem for the patient [6]. Each person receives a score on the "test", the higher the score, the greater the perceived level of pathology (or level of functional ability in the case of VISA-A). The validity of the PROM (i.e., whether it actually measures what it purports to measure) depends directly on the relevance and comprehensiveness of the items for the patient group being assessed (content validity) [7] and on whether the response scores to those questions when added together satisfy the basic constraints of measurement (construct validity) [8–10]. When PROM data is congruent with (i.e., 'fits') certain statistical measurement models, such as item response theory (IRT) or confirmatory factor analysis (CFA), it follows that the PROM possesses adequate psychometric properties [11–13].

As mentioned, significant flaws in the validity of VISA-A were found using the COSMIN checklist [4]. Considering the harsh judgement passed by the COSMIN review, we believe it could be relevant to evaluate specifically which methods were used to generate and choose the items in VISA-A, and which methods were originally used to test its psychometric properties. We also believe that it could be of value to directly evaluate the psychometric properties of VISA-A in our own setting (Denmark) using the most stringent methodology. This could give insights into why the COSMIN assessment resulted in such a negative finding.

This study had three major objectives:

1. Evaluate how VISA-A was created and validated by looking at how the items in the original version of VISA-A were generated and chosen for inclusion in the PROM and address content validity

2. Evaluate which statistical methods were used for the psychometric validation of the original VISA-A

3. Conduct a rigorous analysis of the psychometric properties of the local (Danish) version of VISA-A in a cohort of patients with Achilles tendinopathy and healthy controls.

## Methods

First, in order to address the methodological quality of the content validation of the original version of VISA-A, we assessed the methods used to generate and select the items that constitute the body of VISA-A. This included an evaluation of whether the items were generated from the perspective of clinicians or from interviews with patients with respect to the relevance and coverage of the items (content validity) [7]. We also looked at whether the content of the items in the proposed scale were thematically homogenous and whether the response options were logical and easily understood.

Next, we assessed the process used by the creators of VISA-A to confirm the psychometric properties. Thus, we looked at the original methods that were used to evaluate the factor structure and dimensionality of VISA-A, whether there was evidence of fit to an appropriate measurement model, and whether the authors assessed differential item functioning (DIF). DIF is the presence of bias due to different response patterns in specific items between subgroups, such as sex, age group, or injury chronicity [14–17]. DIF is detrimental to scale properties

since it can mask real differences, or detect differences between subgroups, that are not due to real changes [16]. In such cases, if DIF is present, one cannot discern whether detected differences in VISA-A scores are caused by DIF or real differences in the criterion that VISA-A is assumed to measure. DIF is investigated in models that assess the independence of a list of background variables on the items conditional on the full VISA-A score. Dimensionality is investigated by assessment of data fit to measurement models such as CFA or IRT.

Lastly, we conducted our own analyses of the psychometric properties of VISA-A. These included Rasch IRT, multivariable linear regression, and CFA. The sample was a cohort of 318 persons with Achilles tendinopathy (symptom duration ranging from less than 3 months to more than a year), and a group of 120 healthy persons. The subjects with symptoms < 3 months were sports active participants of both sexes, age 18 or older, with mid-portion Achilles tendon pain recruited from various local sports clubs. The subjects with symptoms > 3 months were 18 to 65 years of age, of both sexes, with mid-portion Achilles tendon pain seen at a sports medicine clinic and a rheumatology outpatient clinic [18]. The healthy persons were male participants, 19 to 90 years, in the 2017 European Masters Athletics Championships [19].

## Analysis strategy

We employed several techniques to assess the psychometric properties of VISA-A. First, fit to a Rasch unidimensional measurement model was assessed using Andersen's conditional likelihood ratio test (CLR) [11]. Overall fit was investigated through obtaining item-trait interaction chi-square values (a non-significant chi-square indicates good fit) [20, 21]. Individual item fit was assessed by standardized individual item-person fit residuals (i.e., the difference between observed and expected scores) to approximate a Z-Score, where values between ±2.5 indicates adequate fit to the model [20, 21]. DIF was assessed using analysis of variance [22] for Sex, Age group (+/- 44 yrs), BMI (+/- 25), and duration of symptoms ($\leq$ 3 months, 4–12 months, $\geq$ 12 months, and no symptoms at all) [16, 23]. For DIF analyses, the cutoff of +/- 44 years of age was chosen because the median age for the sample group was 43.6 years. This allowed for a dichotomization of younger versus older persons for comparison of scoring patterns across the groups. For BMI, the value of 25 was chosen, as this was also the median value for the group. The duration of symptoms groups were chosen to allow for a comparison of scoring patterns across groups that could be expected to have different levels of severity of symptoms (i.e., less than 3 months would be acute symptoms, 4 to 12 months would approach chronic symptoms, and more than 12 months would be manifest chronic tendinopathy).

Due to skewed item response data, which hindered parameter estimation in the Rasch model, we carried out a transformation of the response structure. See the details of this in the results section below.

Next, as the Rasch analysis was performed on transformed data, we used CFA to assess factor structure using the original response data (non-transformed). In these analyses, three separate factor structures of the VISA-A were assessed: the original unidimensional structure, a 2-factor structure (items 1–5 and 6–7), and a 3-factor structure as indicated by the authors in the original paper (items 1–3, 4–6, and 7–8). CFA model fit was assessed with the goodness of fit index (GFI) > 0.95; root mean square error of approximation (RMSEA) < 0.06; standardized root mean square residual (SRMR) < 0.06; and the Comparative Fit Index (CFI) > 0.95 [24, 25].

Lastly, we assessed DIF for the same person characteristics as for the Rasch analyses (Sex, Age group, BMI, and Symptom Duration) in multivariable regression analyses of the individual items, also using the original non-transformed data.

RUMM 2030 was used for the Rasch analysis [23]. CFA and regression analyses were carried out with SAS v9.4. Data for the analyses were accessed from trials conducted at our facility: ClinicalTrials.gov Identifier: NCT03401177 and ClinicalTrials.gov Identifier: NCT02580630. All studies were approved by the local institutional review and ethics committees.

## Results

### Content validity–How was VISA-A developed?

Assessment of the original paper describing the construction of VISA-A [1] revealed that item generation and item reduction was based on literature review and clinician consensus with little or no patient involvement. Further scrutiny showed that each item possesses a mix of themes addressing stiffness, pain, and perceived level of ability, both within individual items and across the 8-item scale. The reader is referred elsewhere for a formal presentation of the items and response structure in VISA-A [1]. However, item 3 is an example of a complicated item concerning pain within the next 2 hours after walking on flat ground for 30 minutes. Is the person scoring pain, the ability to walk on flat ground for 30 minutes, or the ability to walk at all due to pain for 2 hours after having walked for 30 minutes? Items 6, 7, and 8 are also complex with complicated scoring options and thematic ambiguity. Most notable is item 8, which has a mutually exclusive "either. . .or" scoring structure that crosses categories of pain/ no pain with level of training ability. Intuitively, patients may be confused and uncertain as to which theme they are responding. Such items are known as 'double barreled items' or 'ambiguous' [26].

Moreover, as VISA-A is scored as a single index (total score); this assumes that all items address unique aspects of the same overall construct (a single dimension). However, closer inspection reveals that the item content addresses both symptoms, activity level, and activity duration (which are separate constructs). Moreover, the items ask about less demanding functional activities (items 1–5), and more demanding sports-related activities (items 6–8), which potentially are different situational contexts. Indeed, in the original paper by Robinson et al. [1], they mention that VISA-A covers three separate domains of pain, function, and activity, which indicates an underlying multidimensional structure that would not support calculating a singular total score.

In terms of response structure, items 1–6 use a 0–10 numeric rating scale with 11 response options, instead of adjectival response scales, as is more typical for PROMs [7]. Items 7 and 8 do have a 4-option categorical structure, which is transformed to 0–10 rating scale (probably in order to fit in with the other items). However, the result is that items 1–7 can achieve up to 70 points, while item 8 has a max score of 30 points. Thus, the VISA-A can tally a maximum score of 100, although there is no obvious reason for assigning item 8 three times the weight of the others. Nor is there an explanation or description of how and why the clinician-based focus groups chose to include the selected eight items (and thus exclude other potential items). This process does not satisfy the general principles of establishing content validity, which requires face-to-face cognitive interviews with the targeted patients to confirm both the relevance, coverage, and understandability of the items and response options [7, 26].

An additional issue with item 7 is that this is the only item where it is explicitly assumed that the person has symptoms. Hence, in a strict sense, item 7 is not relevant for people without Achilles tendon symptoms, which is problematic if healthy persons were used in the validation process (as was the case for VISA-A). An alternative wording such as "6 months ago" instead of "when the symptoms started" might remedy this. In addition, a max score on item 7 can only be achieved for competitive athletes, whereas recreational athletes with high-volume

training, who do not participate in competitions, receive a lower score regardless of their actual level of functional ability. These are issues that relate directly to item relevance and comprehensiveness in VISA-A.

## Construct validation–How was VISA-A validated?

To test construct validity, the creators of VISA-A calculated Spearman correlation coefficients between VISA-A and two legacy PROMs (i.e., the Percy-Conochie and Curwin-Stanish scales) in a group of non-surgical patients (n = 45), a pre-surgical group (n = 14), and 83 healthy persons. The psychometric properties (i.e., whether VISA-A behaves as a proper measure) were not investigated. This is highly unfortunate, as analysis of these properties form the core of construct validation, notably the most problematic violations of these properties: multidimensionality and differential item functioning (DIF). Ignoring multidimensionality can at best induce variance and make for a weak instrument, or worst case, when the dimensions engage in a trade-off, make for a meaningless instrument.

## Psychometric analyses of the Danish VISA-A

Table 1 shows the characteristics of the people in the sample, the variables used for the DIF analyses, and the VISA-A total scores across subgroups.

**Rasch analysis.** Fit to a partial-credit Rasch model was attempted, but initial model estimation failed due to ceiling effect in the item response scales for items 1 through 5. An example of this is seen in Fig 1, which exhibits the frequency distribution of response scores on item 3 for patients with symptoms lasting 3 months or less. The failed parameter estimation was also likely due to excessive response categories, with 10 category probability thresholds to be estimated for each item, with the exception of item 7 with 4 categories and thus 3 thresholds, which yields 73 threshold estimates for all items combined.

To remedy this, the 0–10 response scales were recoded into four categories, matching the response structure of item 7. The recoding was: (0–2 = 0), (3–5 = 1), (6–8 = 2), and (9–10 = 3). A 4-category response scale was then established for all items, which resulted in 24 thresholds and successful model estimation. Table 2 shows that overall fit to the Rasch model was rejected for the combined item set (significant chi-square). Individually, items 2, 3, 5, and 7 exhibited misfit and DIF was observed for Sex in item 3 and for Duration Group in items 3, 5, 6, 7, and 8 (Table 2). DIF was not observed for BMI or age group. Splitting items for DIF [16] for Sex and Duration Group did not remedy model fit (these results not shown).

**Table 1. Demographic variables for DIF analyses and total VISA-A scores.**

|  |  | N (%) | Total Score Mean (SD) |
|---|---|---|---|
| **Sex** | Male | 333 (76.0) | 74,64 (22.4) |
|  | Female | 105 (24.0) | 52.69 (21.1) |
| **Duration (months)** | ≤3 | 220 (50.2) | 63.99 (16.4) |
|  | 4–12 | 51 (11.6) | 45.47 (18.9) |
|  | >12 | 47 (10.7) | 46.33 (16.6) |
|  | No symptoms | 120 (27.4) | 97.63 (6.9) |
| **Body-mass index (BMI)** | ≤25 | 248 (56.8) | 73.66 (24.3) |
|  | >25 | 189 (43.2) | 63.58 (22.3) |
| **Age (years)** | ≤43 | 180 (41.1) | 64.97 (21.2) |
|  | >43 | 258 (58.9) | 72.48 (25.4) |

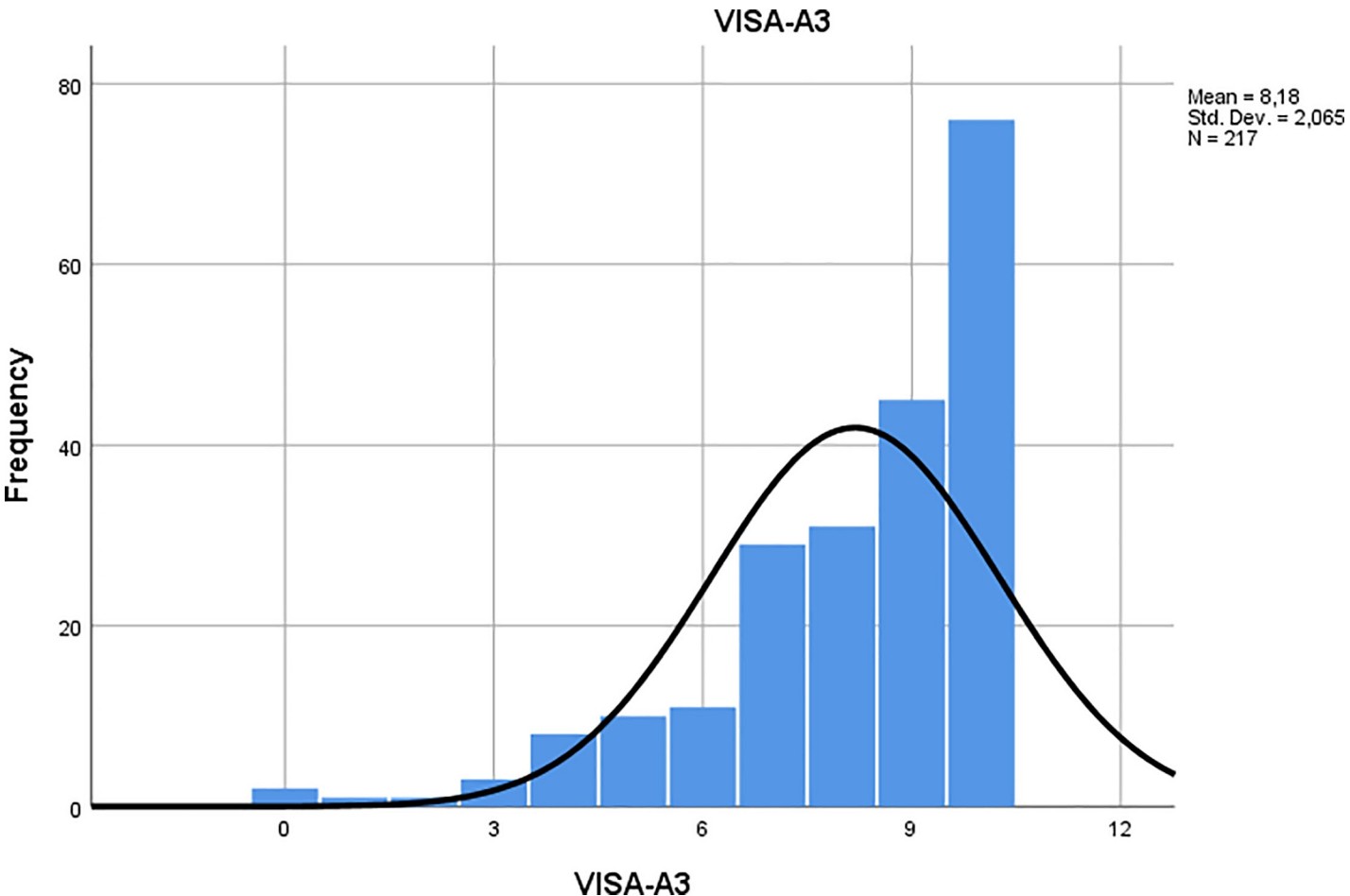

**Fig 1. Response frequency for item 3.** Negative skew for group 1 (symptoms ≤ 3 months).

**Confirmatory factor analysis (CFA).** As the Rasch analysis was carried out on transformed data, thus ignoring the specific scoring and weighting of the items inherent to VISA-A, we used the original scoring of the items for the CFA analysis and multivariable regression. Consistent with the Rasch results, CFA rejected a unidimensional scale and confirmed a

**Table 2. Individual item fit, DIF for sex and symptom duration, and overall fit to the Rasch model.**

|  | Item fit | | DIF for sex | | | DIF for symptom duration | | | Overall fit |
|---|---|---|---|---|---|---|---|---|---|
|  | $X^2$ | P | MS | F | p | MS | F | p | Total Item Chi Squ 115.727 Total Deg of Freedom 48.000 Total Chi Squ Prob **0.000***|
| **Item 1** | 8.517 | 0.203 | 0.67 | 0.764 | 0.382 | 0.74 | 0.820 | 0.483 | |
| **Item 2** | 18.925 | **0.004*** | 0.05 | 0.072 | 0.788 | 2.35 | 3.489 | **0.016*** | |
| **Item 3** | 20.713 | **0.002*** | 9.34 | 16.187 | **0.000*** | 11.72 | 23.048 | **0.000*** | |
| **Item 4** | 6.089 | 0.413 | 0.24 | 0.299 | 0.584 | 1.10 | 1.359 | 0.255 | RELIABILITY INDICES CRONBACH Alpha With Extm: 0.793 NO Extm: 0.779 |
| **Item 5** | 15.514 | **0.020*** | 1.15 | 1.841 | 0.175 | 1.25 | 2.110 | 0.098 | |
| **Item 6** | 4.368 | 0.627 | 0.00 | 0.003 | 0.953 | 6.05 | 6.792 | **0.000*** | |
| **Item 7** | 34.035 | **0.000*** | 3.82 | 3.937 | **0.048*** | 19.55 | 23.460 | **0.000*** | |
| **Item 8** | 7.567 | 0.271 | 0.33 | 0.341 | 0.559 | 8.89 | 10.299 | **0.000*** | |

$X^2$ = Chi Sq, MS = Mean Sq. (ANOVA), F = F-statistic. An asterisk indicates a significant result at the 5% level.

**Table 3. Results of confirmatory factor analyses (CFA).**

|  | GFI | RMSEA | SRMR | CFI | Chronbach's a |
|---|---|---|---|---|---|
| **Target value** | >0.95 | <0.06 | <0.06 | >0.95 | >0.95 |
| **1 dimension** | 0.8865 | 0.1452 | 0.0605 | 0.9088 | 0.804 |
| **2 dimensions** | 0.9376 | 0.0983 | 0.0497 | 0.9603 | 0.892 \| 0.663 |
| **3 dimensions** | 0.9614 | 0.0830 | 0.0401 | 0.9747 | 0.824 \| 0.773 \| 0.576 |

**GFI** = Goodness of Fit; **RMSEA** = Root Mean Square Error of Approximation; **SRMR** = Standardized Root Mean Residual; **CFI** = Comparative Fit Index.

multidimensional structure with items 1–5 in one dimension and items 6–8 in the other (i.e., a 2-factor solution). CFA indicated even more strongly a 3-factor structure with items 1–3, 4–6, and 7–8 in separate dimensions. The results are seen in Table 3.

**Multivariable regression.** Multivariable regression analyses confirmed DIF for sex in item 3 and DIF across duration of symptoms for most items (also seen in the Rasch analysis). Table 4 shows the full DIF results. This DIF persisted in a sensitivity analysis where the 120 participants in the 2017 European Masters Athletics Championships were omitted (results not shown).

## Discussion

These results support the findings of the COSMIN review conducted by Ortega-Avila and colleagues [4], in that we could confirm significant flaws in the validity of VISA-A. As a PROM, VISA-A clearly lacks content validity, as patients were not included in the process of item generation or item reduction, the adequacy of the measurement properties of VISA-A was never confirmed using appropriate validation methods, and our own rigorous analyses using data from patents with Achilles tendinopathy revealed substantial problems. Without patient feedback to generate item content, how do we know if relevant and understandable questions are being asked? Does the scoring structure of each item make sense for patients? For example, what does it actually mean for a patient to score a 3 or an 8 on item 3? Trying to determine where to score on an arbitrary pain scale to describe the level of pain that is expected within the next 2 hours after walking 30 minutes is a complex question to answer. The fact that there is extensive ceiling effect for most items (items 1–5) across all duration groups indicates that those items fail to target patients with Achilles tendinopathy adequately. Such a ceiling effect is only ever justifiable for persons without symptoms.

We found that the original validation was superficial. It was based on Spearman correlation of scores from only 59 patients regressed against other legacy PROMs, which themselves may not reflect good measurement of Achilles tendinopathy. This cannot be considered an assessment of the instrument's psychometric properties, but simply a measure of criterion validity, which does not ensure that the criterion instrument used for reference is trustworthy or valid. No tests of dimensionality, fit to a measurement model, or tests of person-item bias (DIF) were conducted.

Our own analyses of these components on a broad sample of patients and healthy persons revealed several problems with the intrinsic measurement properties of VISA-A. First, the assumption of unidimensionality was rejected. Hence, the computation of VISA-A as a total score is problematic. A possible solution here is to divide VISA-A into the two or three subscales that were confirmed using CFA. This is a strategy that can be implemented retrospectively (i.e., pre-existing historic data can still be used to calculate a multidimensional VISA-A score).

**Table 4. Multivariable regression analysis of differential item functioning (DIF) on the covariates sex, duration of symptoms, body mass index (BMI), and age group.**

| | Item 1 DIF (95% CI) | Item 1 p-value | Item 2 DIF (95% CI) | Item 2 p-value | Item 3 DIF (95% CI) | Item 3 p-value | Item 4 DIF (95% CI) | Item 4 p-value | Item 5 DIF (95% CI) | Item 5 p-value | Item 6 DIF (95% CI) | Item 6 p-value | Item 7 DIF (95% CI) | Item 7 p-value | Item 8 DIF (95% CI) | Item 8 p-value |
|---|---|---|---|---|---|---|---|---|---|---|---|---|---|---|---|---|
| **Sex** | | 0.6505 | | 0.3211 | | 0.0004* | | 0.4141 | | 0.7688 | | 0.5983 | | 0.0962 | | 0.6608 |
| Male | ref | | ref | | ref | | ref | | ref | | ref | | ref | | ref | |
| Female | 0.10 (-0.32;0.51) | 0.6505 | 0.19 (-0.19;0.57) | 0.3211 | -0.65 (-1.00;-0.29) | 0.0004* | -0.17 (-0.58;0.24) | 0.4141 | 0.06 (-0.33;0.45) | 0.7688 | -0.17 (-0.81;0.47) | 0.5983 | 0.39 (-0.07;0.85) | 0.0957 | 0.25 (-0.88;1.39) | 0.6608 |
| **Duration (months)** | | 0.1931 | | 0.0142* | | < .0001* | | 0.0507* | | < .0001* | | < .0001* | | < .0001* | | 0.0015* |
| ≤ 3 | ref | | ref | | ref | | ref | | ref | | ref | | ref | | ref | |
| 4–12 | 0.02 (-0.53;0.57) | 0.9412 | 0.52 (0.01;1.03) | 0.0441* | -0.56 (-1.03;-0.08) | 0.0214* | 0.66 (0.11;1.21) | 0.0187* | 1.09 (0.57;1.61) | < .0001* | 2.07 (1.21;2.93) | < .0001* | -0.96 (-1.57;-0.34) | 0.0034* | -2.85 (-4.36;-1.33) | 0.0002* |
| ≥ 12 | -0.51 (-1.08;0.06) | 0.0794 | 0.48 (-0.05;1.00) | 0.0761 | -1.68 (-2.17;-1.19) | < .0001* | 0.42 (-0.15;0.99) | 0.1455 | 1.17 (0.63;1.71) | < .0001* | 2.05 (1.17;2.94) | < .0001* | -1.80 (-2.44;-1.16) | < .0001* | -0.13 (-1.70;1.45) | 0.8719 |
| No Sympt. | 0.59 (-0.47;0.21) | 0.2716 | 1.17 (0.20;2.14) | 0.0178* | 0.25 (-0.65;1.15) | 0.5868 | 0.75 (-0.29;1.80) | 0.1581 | 0.33 (-0.66;1.32) | 0.5192 | 0.85 (-0.79;2.48) | 0.3102 | -5.44 (-6.62;-4.27) | < .0001* | 1.50 (-1.39;4.40) | 0.3086 |
| **BMI (kg/m2)** | | 0.4458 | | 0.8257 | | 0.8411 | | 0.9039 | | 0.2192 | | 0.0895 | | 0.7520 | | 0.5266 |
| ≤ 25 | ref | | ref | | ref | | ref | | ref | | ref | | ref | | ref | |
| > 25 | -0.13 (-0.47;0.21) | 0.4458 | 0.04 (-0.28;0.35) | 0.8257 | -0.03 (-0.32;0.26) | 0.8410 | 0.02 (-0.32;0.36) | 0.9039 | 0.20 (-0.12;0.52) | 0.2188 | -0.46 (-0.99;0.07) | 0.0895 | 0.06 (-0.32;0.44) | 0.7520 | 0.30 (-0.63;1.24) | 0.5265 |
| **Age (years)** | | 0.2542 | | 0.3048 | | 0.6065 | | 0.7579 | | 0.9333 | | 0.9573 | | 0.3392 | | 0.9243 |
| ≤43 | ref | | ref | | ref | | ref | | ref | | ref | | ref | | ref | |
| ≥44 | 0.20 (-0.14;0.53) | 0.2542 | 0.16 (-0.15;0.47) | 0.3045 | -0.08 (-0.37;0.21) | 0.6064 | -0.05 (-0.39;0.28) | 0.7579 | 0.01 (-0.30;0.33) | 0.9333 | -0.01 (-0.54;0.51) | 0.9573 | -0.18 (-0.56;0.19) | 0.7579 | -0.05 (-0.97;0.88) | 0.9243 |

The results show DIF for sex for item 3 and for symptom duration for all items except item 1. An asterisk indicates a significant result at the 5% level.

Probably the greatest problem was that there was DIF for the covariate 'duration of symptoms' across all but one item, which suggests that VISA-A measures a different construct for patients in the different symptom duration groups. When the construct being measured changes over time, the meaning of intervention effects that reach over longer periods is undermined, particularly if the groups compared are defined by the duration of symptoms. Hence, to neutralize this DIF, we suggest that if VISA-A is used as outcome, conducting trials with follow-up longer than three months should be avoided, and only comparison of patients that all have the same (short) duration of symptoms at baseline should be undertaken. This is important because comparisons of constructs that change over time or between groups will undermine the interpretation of intervention effects.

In contrast with our results, other studies have found VISA-A to be valid and reliable [27]. However, a closer review of those reports reveals that the validation methods closely mirror those used in the original paper [28, 29], which means they fail to satisfy the basic constraints of content validity and the psychometric measurement properties. There are two notable exceptions. One group [30] found a found a 2-factor structure for items 1–6 and 7–8 using exploratory factor analysis, but with only 51 patients, and the fact that confirmatory tests were never performed, the results cannot be considered robust (although they somewhat agree with our findings). A more recent study concluded that a 1-factor solution was viable using CFA [29]. However, the analysis was based on data from just 70 patients, and the study unfortunately did not assess measurement invariance (DIF).

We failed to generate a Rasch model for the proposed 8-item construct with the 11-category visual analog response scales. We therefore restructured the response scales, which allowed for successful parameter estimation. However, this still revealed substantial misfit and DIF. In order to accommodate the original scoring structure of the VISA-A, we conducted analyses of dimensionality and DIF in the original format using CFA and multivariable regression. Here, we found no major differences between the results of the Rasch analysis, the CFA, and the linear multivariable analyses, and therefore feel justified in our choice of methods.

Our results support the findings of Ortega-Avila and colleagues [4], in which which they used the COSMIN checklist and found significant flaws in the construct validity of VISA-A. We chose not to apply COSMIN for our analyses. First, because it would have been redundant, as Ortega-Avila et al. included the Danish version in their study, and second, while COSMIN is an exhaustive tool for assessing which methods have been used to create and validate PROMs, it does not specifically address the superiority of one validation method relative to others. For example, COSMIN does not consider whether CFA or IRT is more (or less) robust than for example exploratory factor analysis (EFA), or correlation with legacy instruments (criterion validation). Therefore, we applied the most robust assessment methods to assess the psychometric properties, as we found no studies that previously had applied Rasch IRT, CFA, or the multivariable analyses we chose to use. Moreover, while Ortega-Avila et al. specifically targeted the 11 studies in the different language versions of VISA-A that assessed the construct validity and measurement characteristics, we focused more on the process behind the genesis and the validation of the original PROM and sought to verify these results with our own analyses.

## Conclusion

VISA-A is not a robust scale for measuring Achilles tendinopathy. It lacks content validity and construct validity, and thorough validation methods were not used to test its measurement properties during the development phase or subsequently thereafter. Furthermore, rigorous psychometric assessment of the Danish version revealed that VISA-A does not satisfy a

measurement model, lacks unidimensionality, and exhibits DIF depending on the duration period of symptoms. A new relevant PROM for Achilles tendinopathy should be developed and appropriately tested for validity. Meanwhile, simple pain scoring (e.g., numeric rating scales) and functional tests are suggested as more appropriate outcome measures for studies of Achilles tendinopathy. VISA-A sub-scores can still be calculated as described in the original paper, which means that existing research using VISA-A data need not be discarded. However, this option does not address the poor psychometric properties of VISA-A.

## Supporting information

**S1 File.**
(SAV)

## Author Contributions

**Conceptualization:** Jonathan Comins, Volkert Siersma, Christian Couppe, Rene B. Svensson, Finn Johansen, S. Peter Magnusson.

**Data curation:** Jonathan Comins, Christian Couppe, Rene B. Svensson, Finn Johansen, Nikolaj M. Malmgaard-Clausen, S. Peter Magnusson.

**Formal analysis:** Jonathan Comins, Volkert Siersma, Rene B. Svensson, S. Peter Magnusson.

**Investigation:** Finn Johansen, Nikolaj M. Malmgaard-Clausen.

**Methodology:** Jonathan Comins, Volkert Siersma, Christian Couppe.

**Project administration:** S. Peter Magnusson.

**Resources:** Christian Couppe, Finn Johansen, S. Peter Magnusson.

**Validation:** Jonathan Comins, Volkert Siersma.

**Writing – original draft:** Jonathan Comins, Volkert Siersma, Christian Couppe, Rene B. Svensson, Finn Johansen, Nikolaj M. Malmgaard-Clausen, S. Peter Magnusson.

**Writing – review & editing:** Jonathan Comins, Volkert Siersma, Christian Couppe, Rene B. Svensson, Finn Johansen, Nikolaj M. Malmgaard-Clausen, S. Peter Magnusson.

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
