## [Editor Report · Decision Letter 0]

28 Aug 2020

PONE-D-20-08074

Assessment of content validity and psychometric properties of VISA-A for Achilles tendinopathy.

PLOS ONE

Dear Dr. Comins,

Thank you for submitting your manuscript to PLOS ONE. After careful consideration, we feel that it has merit but does not fully meet PLOS ONE’s publication criteria as it currently stands. Therefore, we invite you to submit a revised version of the manuscript that addresses the points raised during the review process.

We look forward to receiving your revised manuscript.

Kind regards,

Alison Rushton

Academic Editor

PLOS ONE

Additional Editor Comments:

Please amend the methods section as we have discussed through email to ensure the rigour and detail of the methodology is communicated.

Specifically, the manuscript is currently lacking adherence to methodological guidelines (COSMIN) and the methodology section is under-developed, consisting of only 7 lines.

Please revise the manuscript to further develop communication of the methodological rigor of the study and submit a revised version. I recognize that some methodological points are detailed through out the results section, but please collate them into a robust methods section.

The following resource will assist you in further developing the manuscript:

https://www.cosmin.nl/wp-content/uploads/COSMIN-study-designing-checklist_final.pdf

Journal Requirements:

2. In methods section please clarify whether you obtained IRB approval and participant informed consent for the original studies from which you extracted data for the current article.

---

## [Decision Letter · Decision Letter 1]

5 Jan 2021

PONE-D-20-08074R1

Assessment of content validity and psychometric properties of VISA-A for Achilles tendinopathy.

PLOS ONE

Dear Dr. Comins,

Thank you for submitting your manuscript to PLOS ONE. After careful consideration, we feel that it has merit but does not fully meet PLOS ONE’s publication criteria as it currently stands. Therefore, we invite you to submit a revised version of the manuscript that addresses the points raised during the review process.

Thank you for submitting this interesting manuscript that has received very positive reviews.

Please address the minor comments raised by reviewers.

We look forward to receiving your revised manuscript.

Kind regards,

Alison Rushton

Academic Editor

PLOS ONE

Reviewers' comments:

Reviewer's Responses to Questions

**Comments to the Author**

1. If the authors have adequately addressed your comments raised in a previous round of review and you feel that this manuscript is now acceptable for publication, you may indicate that here to bypass the “Comments to the Author” section, enter your conflict of interest statement in the “Confidential to Editor” section, and submit your "Accept" recommendation.

Reviewer #1: (No Response)

Reviewer #2: (No Response)

Reviewer #3: All comments have been addressed

2. Is the manuscript technically sound, and do the data support the conclusions?

Reviewer #1: Yes

Reviewer #2: Partly

Reviewer #3: Yes

3. Has the statistical analysis been performed appropriately and rigorously? 

Reviewer #1: Yes

Reviewer #2: Yes

Reviewer #3: I Don't Know

4. Have the authors made all data underlying the findings in their manuscript fully available?

Reviewer #1: Yes

Reviewer #2: Yes

Reviewer #3: Yes

5. Is the manuscript presented in an intelligible fashion and written in standard English?

Reviewer #1: Yes

Reviewer #2: Yes

Reviewer #3: Yes

6. Review Comments to the Author

Reviewer #1: The manuscript addresses an important area, the PROM instrument validity. As a statistician in outcomes research, the PROM is gaining popularity. However, people rarely go back to review or revalidate the instrument. This manuscript layout a clear pathway and evaluation metrics on how to re-evaluate the instrument for intended population and can be applied more routinely in PROM development.

The manuscript is clearly written and results are displayed thoughtfully.

Reviewer #2: Congratulations with such a great work. I would like to comment on a few points to make this paper flow better:

1. (Line 119-120) Can the author provide citation or explain the reason(s) of having such cut-off point for age, BMI and duration of symptoms?

2. (Table 4) Would suggest to standardize the decimal places for the numbers. In addition, suggest to keep to one p-value, unless there's multiple category (e.g. in the sex category, the same p-value appeared twice)

3. (Line 251) Suggest to standardize the numbers in one sentence (i.e. eight, three and 3)

4. Based on my understanding, VISA-A questionnaire measures clinical severity of Achilles tendinopathy. Throughout the paper, it was not clearly mentioned about what VISA-A measures (mentioned in the paper "VISA-A measures Achilles tendinopathy, which is slightly confusing). Perhaps, the authors could clearly explain it in the introduction and conclusion, so that the readers are clearer.

Reviewer #3: I have not been part of the first review. I like the reviewed paper and suggest accepting it in its new form.

7. PLOS authors have the option to publish the peer review history of their article (what does this mean?). If published, this will include your full peer review and any attached files.

Reviewer #1: No

Reviewer #2: No

Reviewer #3: No

---

## [Author Response · Author response to Decision Letter 1]

5 Jan 2021

Dear Editor,

We have addressed the comments by reviewer 2 and hope that our responses are acceptable. Thank you for the chance to clarify. 

Jonathan Comins

---

## [Editor Report · Decision Letter 2]

3 Feb 2021

Assessment of content validity and psychometric properties of VISA-A for Achilles tendinopathy.

PONE-D-20-08074R2

Dear Dr. Comins,

We’re pleased to inform you that your manuscript has been judged scientifically suitable for publication and will be formally accepted for publication once it meets all outstanding technical requirements.

Kind regards,

Alison Rushton

Academic Editor

PLOS ONE

Additional Editor Comments (optional):

Thank you for addressing all reviewers' comments to a satisfactory level.

As you are aware, we also invited the corresponding author of the Robinson et al. paper (https://bjsm.bmj.com/content/35/5/335) to provide a review of your study, as per our policy on manuscripts disputing published work (https://journals.plos.org/plosone/s/submission-guidelines#loc-manuscripts-disputing-published-work). The corresponding author has declined our invitation to review.

---

## [Editor Report · Acceptance letter]

2 Mar 2021

PONE-D-20-08074R2 

Assessment of content validity and psychometric properties of VISA-A for Achilles tendinopathy. 

Dear Dr. Comins:

I'm pleased to inform you that your manuscript has been deemed suitable for publication in PLOS ONE. Congratulations! Your manuscript is now with our production department. 

Kind regards, 

on behalf of

Professor Alison Rushton 

Academic Editor

PLOS ONE